# Long QT syndrome type 3 gain-of-function of Na$_v$1.5 increases ventricular fibroblasts proliferation and pro-fibrotic factors
Claire Castro [1,2], Justine Patin[1], Cyrielle Jajkiewicz [1], Franck Chizelle[1], Cynthia Ore Cerpa[1], Agnès Tessier [1], Eva Le Pogam[1], Imen Fellah[1], Isabelle Baró [1], Flavien Charpentier[1] & Mickaël Derangeon [1] ✉

The long QT syndrome type 3 (LQT3) is a cardiac channelopathy caused by gain-of-function mutations in the *SCN5A* gene, encoding the sodium channel Na$_v$1.5. As Na$_v$1.5 is expressed in cardiomyocytes but also in cardiac fibroblasts, we investigated whether the LQT3-causing p.ΔQKP1507-1509 (ΔQKP) *SCN5A* mutation alters cardiac fibroblast phenotype. Primary cultured ventricular fibroblasts from *Scn5a*$^{+/ΔQKP}$ knock-in mice showed increased proliferation, survival, expression of transforming growth factor-β (TGF-β) and activation of its canonical pathway, and reduced α-smooth muscle actin expression. Ventricular tissue from *Scn5a*$^{+/ΔQKP}$ mice exhibited augmented fibroblast populations and fibrosis. Inhibiting TGF-β receptor, sodium current or *Scn5a* expression decreased *Scn5a*$^{+/ΔQKP}$ fibroblast proliferation, while veratridine increased proliferation of control fibroblasts, mimicking Na$_v$1.5 gain-of-function. Lastly, abnormal calcium signaling underlied the increased proliferation of *Scn5a*$^{+/ΔQKP}$ fibroblasts. Our study shows that cardiac fibroblasts carrying the ΔQKP-*SCN5A* mutation exhibit an abnormal, proliferative phenotype, paving the way for better understanding the role of cardiac fibroblasts in LQT3.

The congenital long QT syndrome type 3 (LQT3) is an inherited cardiac arrhythmia disease characterized by an alteration of ventricular repolarization leading to a prolongation of the QT interval on the electrocardiogram (ECG). It is associated with a characteristic polymorphic ventricular tachycardia, *i.e.*, *torsades de pointes*, ventricular fibrillation, syncope and sudden cardiac death[1]. LQT3 is caused by mutations of *SCN5A* gene, encoding the voltage-gated Na$^+$ channel Na$_v$1.5. Na$_v$1.5 mediates the action potential upstroke in cardiomyocytes and is therefore crucial for cardiac excitability and electrical propagation. Commonly, LQT3-causing mutations lead to a delay in Na$^+$ current inactivation, and/or the occurrence of a non-inactivating persistent component of the Na$^+$ current resulting in action potential prolongation[2].

For some LQT3 patients such as patients carrying the QKP1507-1509 amino-acids deletion[3,4], electrical abnormalities are associated with dilated cardiomyopathy [3]. In order to understand the link between the QT prolongation and the cardiomyopathy, our team has developed a knock-in mouse model carrying the equivalent delQKP1510-1512 mutation (*Scn5a*$^{+/ΔQKP}$ mouse). We have previously showed that the *Scn5a*$^{+/ΔQKP}$ mouse presents an abnormally large late Na$^+$ current leading to prolonged QT interval,

ventricular arrhythmias and secondary calcium signaling changes, cardiomyocytes hypertrophy and fibrosis[5], thus mimicking the human phenotype.

Chu and collaborators have showed the implication of ventricular fibroblasts in the development of a drug-induced Long QT syndrome (LQTS) by demonstrating that transforming growth factor β (TGF-β) produced by fibroblasts induced a down-regulation of Kv11.1 (hERG) and Kir2.1 K$^+$ channels expression in cardiomyocytes[6]. Both channels generate K$^+$ currents involved in action potential repolarization and contribute to the cardiac repolarization reserve. Reduced repolarization reserve makes it more likely that further added stress of, for example, a hERG-blocking drug, is sufficient to precipitate arrhythmias such as *torsades de pointes*. Cardiac fibroblasts are an abundant cell type in heart[7] and are now recognized as an important actor implicated in cardiac pathophysiology. Indeed, they regulate the homeostasis of the extracellular matrix and are essential to the communication between cells through direct interactions and/or production of paracrine factors[8]. Cardiac fibroblasts can also modulate electrophysiological processes by direct and indirect interaction with cardiomyocytes[9,10]. Interestingly, it has been shown that Na$_v$1.5 is also expressed in cardiac fibroblasts[11–14]. However, the consequences of LQT3

[1]Nantes Université, CHU Nantes, CNRS, INSERM, l'institut du thorax, F-44000 Nantes, France. [2]Present address: Corrigan Minehan Heart Center, Massachusetts General Hospital, Harvard Medical School, Boston, MA, 02114, USA. ✉e-mail: mickael.derangeon@univ-nantes.fr

*SCN5A* mutations on cardiac fibroblasts and their involvement in LQT3 pathophysiology are unknown.

To clarify these points, we investigated the phenotype of cultured ventricular fibroblasts from *Scn5a*$^{+/\Delta QKP}$ mice. We show that ventricular fibroblasts derived from the LQT3 mouse model exhibit a higher proliferation rate and cell survival than fibroblasts from control mice, i.e., wild-type (*Scn5a*$^{+/+}$) and *Scn5a*$^{+/+}$-*flp* mice (*cf.* Methods section). Furthermore, these fibroblasts display a reduced capacity to differentiate into myofibroblasts, although they exhibit elevated expression of TGF-β and an activation of TGF-β canonical pathway, resulting in increased expression of type I and type III collagens.

## Results

### Ventricular fibroblasts abundance is increased in the hearts of *Scn5a*$^{+/\Delta QKP}$ mice

As previously observed[5], hearts from 4-week-old *Scn5a*$^{+/\Delta QKP}$ mice were bigger than those from control mice (Fig. 1a). This increase in heart size was attributed to increased left ventricular wall and septum thickness due, at least in part, to cardiomyocyte hypertrophy[5]. In this study, we investigated the relative quantity of ventricular fibroblasts. As shown in Fig. 1b, the area of cardiac sections occupied by fibroblasts in *Scn5a*$^{+/\Delta QKP}$ hearts was significantly larger than in hearts from control mice ($P = 0.0007$). In addition, we observed that ventricular fibroblasts from *Scn5a*$^{+/\Delta QKP}$ mice were significantly more abundant than fibroblasts from control mice at day 5 of culture ($P < 0.0001$; Fig. 1c). Finally, ventricular expression of vimentin, a marker of non-myocyte cells, was significantly larger, at the protein level, in *Scn5a*$^{+/\Delta QKP}$ hearts ($P < 0.0001$; Fig. 1d; Supplementary Fig. 1). However, similar vimentin expression was measured in isolated fibroblasts from the two mouse types (Fig. 1d), thus confirming the increase in fibroblast abundance in *Scn5a*$^{+/\Delta QKP}$ hearts at the age of 4 weeks. In contrast, in cardiac sections from 2-week-old mice, fibroblast abundance did not differ between *Scn5a*$^{+/\Delta QKP}$ and control mice, as shown by vimentin staining experiments (Supplementary Fig. 2). However, the slight non-significant increase in cell

proliferation, as shown by Ki67 signal measurement ($P = 0.0589$; Supplementary Fig. 2), suggests that the age of two weeks corresponds to the onset of fibroblast proliferation.

### *Scn5a*$^{+/\Delta QKP}$ ventricular fibroblasts proliferate and survive more than control fibroblasts

In order to identify the mechanisms for the increased abundance of ventricular fibroblasts in *Scn5a*$^{+/\Delta QKP}$ mice, we investigated their proliferation with the xCELLigence technique. As shown in Fig. 2a, the proliferation rate of 4-week-old mice *Scn5a*$^{+/\Delta QKP}$ ventricular fibroblasts in culture was higher than for control fibroblasts ($P < 0.0001$). Interestingly, immunostaining experiments using Ki67 as a marker of proliferation on ventricular sections showed that the number of cells in division was significantly larger in *Scn5a*$^{+/\Delta QKP}$ mice than in control mice ($P = 0.0136$; Fig. 2b). Together, these results suggest that, in vivo, fibroblasts proliferate more in *Scn5a*$^{+/\Delta QKP}$ mice than in control mice. Indeed, cardiomyocytes are well-known to not proliferate at this age and our results on CD68 cardiac expression (Supplementary Fig. 3) suggest that the quantity of macrophages does not differ between control and *Scn5a*$^{+/\Delta QKP}$ mice. In addition, *Scn5a*$^{+/\Delta QKP}$ ventricular fibroblasts presented lower levels of cleaved caspase 3 (marker of apoptosis) than control fibroblasts while total caspase 3 expression was unchanged, reflecting lower apoptosis in *Scn5a*$^{+/\Delta QKP}$ fibroblasts compared to control fibroblasts ($P = 0.0043$; Fig. 2c). This result is reinforced by the observation that *Scn5a*$^{+/\Delta QKP}$ ventricular fibroblasts expressed more Mcl-1, a marker of cell survival, than control fibroblasts ($P = 0.0102$; Fig. 2c). Thus, the higher survival and proliferation of *Scn5a*$^{+/\Delta QKP}$ cardiac fibroblasts suggest that they are activated[15].

### *Scn5a*$^{+/\Delta QKP}$ ventricular fibroblasts express less α-smooth muscle actin (α-SMA) than controls

The expression of α-SMA by cardiac fibroblasts is a marker of late differentiation into myofibroblasts[15]. As shown in Fig. 2d–f, α-SMA was significantly less expressed in *Scn5a*$^{+/\Delta QKP}$ ventricular fibroblasts than in

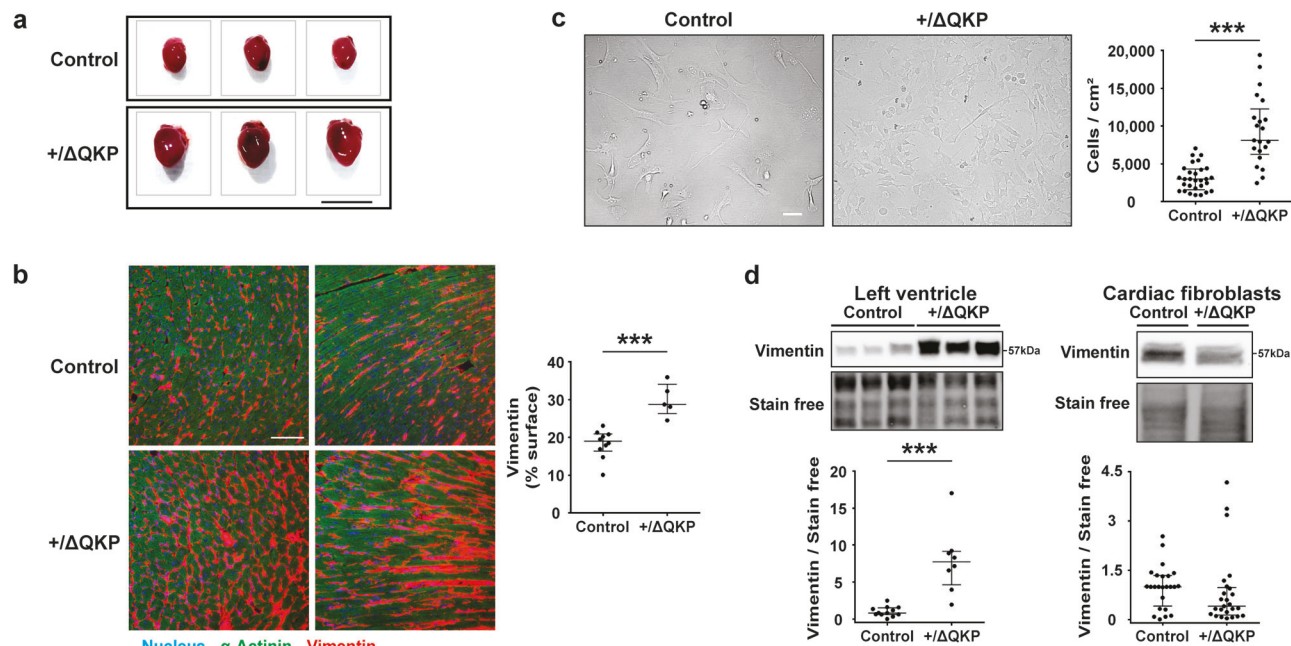

**Fig. 1 | Ventricular fibroblasts abundance is higher in vitro and in vivo in 4-week-old *Scn5a*$^{+/\Delta QKP}$ mice. a** Representative hearts from control and *Scn5a*$^{+/\Delta QKP}$ mice (scale bar: 1 cm). **b** Representative transverse (left panels) and longitudinal (right panels) immunostainings of vimentin (red) in left ventricular sections of control and *Scn5a*$^{+/\Delta QKP}$ mice (scale bar: 50 μm), and vimentin expression (% of cardiac section; $n = 10$ & 5 mice, respectively). Cardiomyocyte marker α-actinin is stained in green. **c.** Representative cultures (scale bar: 50 μm) and cell density (cells/cm²) of fibroblasts

from control and *Scn5a*$^{+/\Delta QKP}$ hearts ($n = 29$ & 21, respectively) at day 5 of culture. **d** Representative western blots and vimentin expression in 4-week-old control and *Scn5a*$^{+/\Delta QKP}$ mouse left ventricles ($n = 12$ & 8, respectively) and fibroblasts at day 8 of culture ($n = 12$ & 14, respectively). The median and interquartile range are shown. The dots indicate the values for each independent mouse. ***$p < 0.001$ (Mann-Whitney test).

**Fig. 2 | Phenotype of ventricular fibroblasts from 4-week-old control and Scn5a$^{+/\Delta QKP}$ mice.**
**a** Representative xCELLigence cell index time course (fibroblasts seeded at 5000 cells/well, 5 days after cell isolation) and proliferation rate between 20 and 25 h post-seeding for fibroblasts from heart of control and Scn5a$^{+/\Delta QKP}$ mice (n = 52 & 38, respectively), normalized to control mean value. **b** Representative immunostainings of Ki67 (red) in left ventricular sections of control and Scn5a$^{+/\Delta QKP}$ mice (scale bar: 50 μm), and the ratio of Ki67-positive nuclei (n = 20 & 11, respectively). **c** Representative western blot and protein expression in fibroblasts from control and Scn5a$^{+/\Delta QKP}$ hearts of cleaved caspase 3 / caspase 3 (n = 14 & 9, respectively), and Mcl-1 (n = 9 & 6, respectively). **d** Representative immunostainings of α-smooth muscle actin (α-SMA, green) in fibroblasts from control and Scn5a$^{+/\Delta QKP}$ hearts, 15 hours post-seeding (scale bar: 100 μm). **e** Acta2 (gene encoding α-SMA) mRNA expression relative to Hprt and normalized to control mean value (2$^{-\Delta\Delta Ct}$; n = 10 & 8, respectively). **f** Representative western blot and protein expression, in cardiac fibroblasts from control and Scn5a$^{+/\Delta QKP}$ mice, of α-SMA (n = 16 & 11, respectively) and periostin (POSTN; n = 21 & 15, respectively). Data obtained from 4-week-old mice. The median and interquartile range are shown. The dots indicate the values for each independent mouse. *, **, ***p < 0.05, 0.01 and 0.001, respectively (Mann-Whitney test).

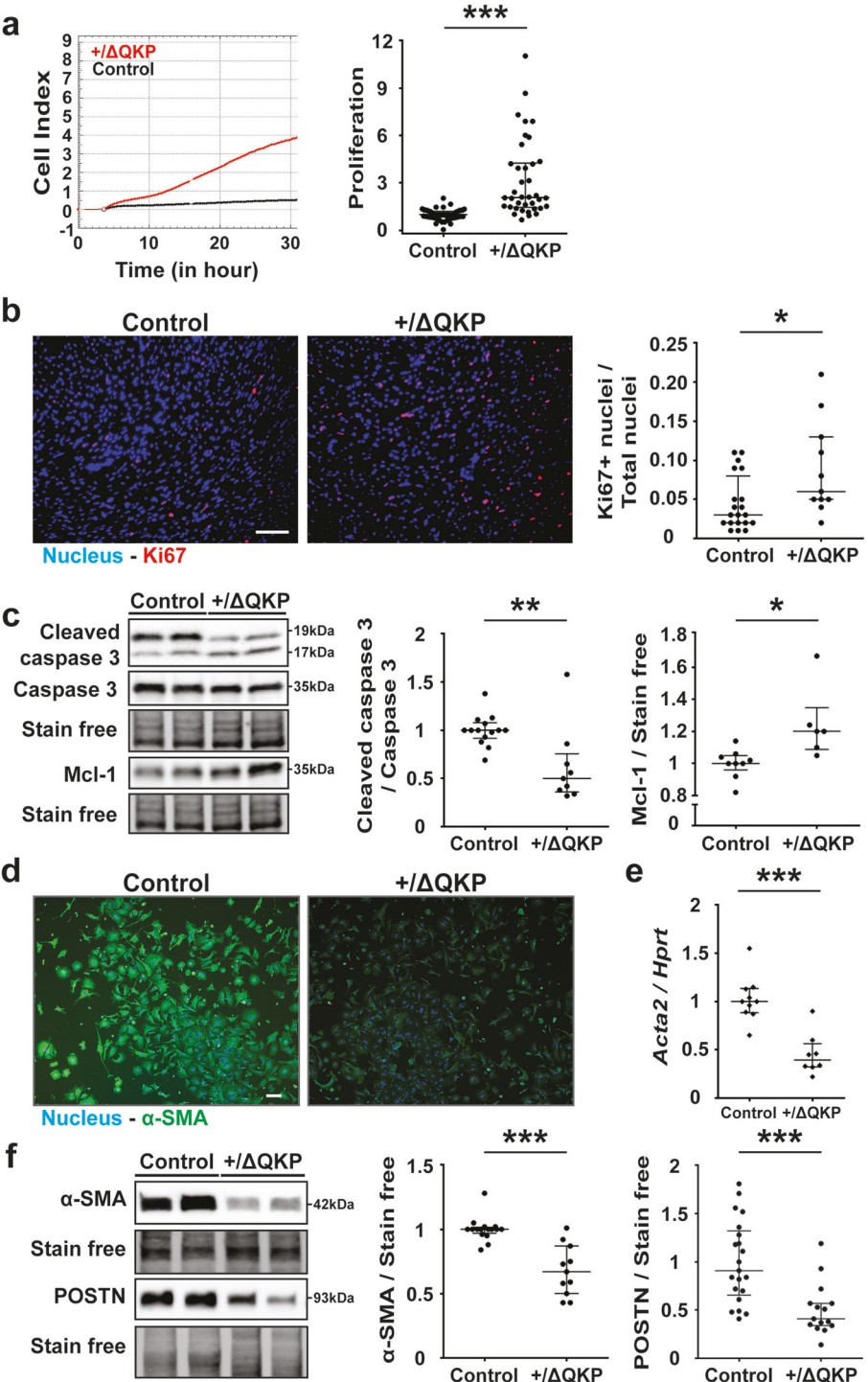

control fibroblasts, at both mRNA (P < 0.0001) and protein levels (P < 0.0001). The expression of periostin, another marker of differentiation, was also lower in Scn5a$^{+/\Delta QKP}$ fibroblasts (P < 0.0001). Therefore, it appears that Scn5a$^{+/\Delta QKP}$ ventricular fibroblasts are less prone to differentiate into myofibroblasts than control fibroblasts. This result was also observed in the right and left ventricles of Scn5a$^{+/\Delta QKP}$ mice (Supplementary Fig. 4).

### Expression of transforming growth factor-β (TGF-β) and activation of the TGF-β canonical pathways in ventricular fibroblasts from Scn5a$^{+/\Delta QKP}$ mice

Depending on their phenotype, the secretion of cardiac fibroblasts can change and impact their environment. As shown in Supplementary Fig. 5a,

mRNA expression of the interleukins Il-1β and Il-6 was higher in Scn5a$^{+/\Delta QKP}$ ventricular fibroblasts compared to control fibroblasts, whereas Il-1α expression was lower. We also observed a possible change in the direct cell-to-cell interaction via gap junctions, and particularly through connexin 43 (Cx43)[10]. Indeed, Cx43 expression in Scn5a$^{+/\Delta QKP}$ ventricular fibroblasts and ventricles were lower than in controls (fibroblasts: P = 0.0001; left ventricle: P = 0.0012, Supplementary Fig. 5b-c). These results suggest changes in fibroblast phenotype that may modify their interactions and impacts on cardiac tissue functions.

Cardiac fibroblast secretion plays a central role in extracellular matrix homeostasis, and is responsible of fibrosis in pathological conditions[8]. Previously, we have shown the development of an interstitial fibrosis in

$Scn5a^{+/\Delta QKP}$ heart[5]. Therefore, we investigated the expression of TGF-β, which is known to be produced by cardiac fibroblasts and involved in fibrosis development[16]. $Scn5a^{+/\Delta QKP}$ ventricular fibroblasts in culture expressed more TGF-β than control fibroblasts at both mRNA ($P < 0.0001$; Supplementary Fig. 6a) and protein ($P = 0.0379$; Fig. 3a) levels, whereas the expression of connective tissue growth factor (CTGF, another profibrotic factor) was unchanged. Furthermore, the TGF-β canonical pathway was activated, as shown by an increase of Smad 2/3 phosphorylation (P-Smad 2/3) in absence of altered expression of total Smad 2/3 ($P = 0.0386$). This activation was associated with a higher production of collagen types I and III (Col I and III; respectively, $P = 0.0069$ and $P = 0.0002$; Fig. 3a). These results were confirmed in ventricular preparations. Interestingly, unlike cultured $Scn5a^{+/\Delta QKP}$ ventricular fibroblasts, $Scn5a^{+/\Delta QKP}$ total ventricles showed a larger CTGF expression than control ones ($P < 0.0001$; Fig. 3b and Supplementary Fig. 6b). Altogether, these results show the involvement of cardiac fibroblasts in fibrosis development, as previously shown[5], and the likely implication of other cell types producing CTGF, which is known to participate to fibrosis development in association with TGF-β[17].

## TGF-β receptor blockade prevents abnormal proliferation of $Scn5a^{+/\Delta QKP}$ ventricular fibroblasts

To determine the involvement of TGF-β in ventricular fibroblast proliferation, these cells were treated by GW788388, a blocker of TGF-β1 receptor. In the presence of GW788388 (20 µM), the proliferation rate of $Scn5a^{+/\Delta QKP}$ ventricular fibroblasts decreased significantly to the levels observed in control fibroblasts ($P = 0.0008$; Fig. 4a). These results suggest that the activation of TGF-β pathway is implicated in the proliferation of ventricular fibroblasts in $Scn5a^{+/\Delta QKP}$ mice.

## Veratridine treatment increases the proliferation of control ventricular fibroblasts

The $Scn5a^{+/\Delta QKP}$ mouse model has a mutation on $Scn5a$ gene that increases the late sodium current generated by $Na_v1.5$ channel. We thus investigated the impact of this current on cardiac fibroblasts. First, we confirmed the expression of $Na_v1.5$ in cardiac fibroblast and specifically in primary ventricular fibroblast (Fig. 4c). Then, we showed that $Scn5a$ gene was expressed at similar mRNA and protein levels in ventricular fibroblasts from $Scn5a^{+/\Delta QKP}$ and control hearts (Supplementary Fig. 7a and Fig. 4c). To mimic the effect of the ΔQKP gain-of-function mutation of $Na_v1.5$, control ventricular fibroblasts were treated with veratridine, which increases the late sodium current. A concentration of 1 µM was chosen so that the late sodium represented 2–3% of the peak current, an effect comparable to the mutation[18,19]. In the presence of veratridine, the proliferation rate of control ventricular fibroblasts was significantly increased by 1.5-fold ($P = 0.0395$; Fig. 4b). Inversely, inhibiting the sodium current with either 50 µM of tetrodotoxin (TTX, a blocker of the peak and late sodium currents) or 50 µM of ranolazine (a preferential blocker of the late sodium current), or decreasing $Scn5a$ expression with 10 nM of an antisense oligonucleotide against $Scn5a$ (ASO) did not change control fibroblast proliferation rate (Supplementary Fig. 7b). These results suggest that WT $Scn5a$ expression and the resulting Na$^+$ current have no impact on proliferation of control ventricular fibroblasts in basal condition, but that an increase of the late sodium current can stimulate their proliferation.

## $Na_v1.5$ inhibition decreases the proliferation of $Scn5a^{+/\Delta QKP}$ ventricular fibroblasts

The same experiments were performed on $Scn5a^{+/\Delta QKP}$ ventricular fibroblasts. Unlike in control fibroblasts, veratridine did not modify the proliferation rate of $Scn5a^{+/\Delta QKP}$ ventricular fibroblasts (Supplementary Fig. 7c), whereas TTX, ranolazine or ASO significantly decreased it, although not to the levels observed in control cardiac fibroblasts (TTX: $P < 0.0001$, ranolazine: $P < 0.0001$, and ASO: $P = 0.0003$; Fig. 4d-e). These results suggest that the late sodium current in $Scn5a^{+/\Delta QKP}$ ventricular fibroblasts is partly responsible for their higher proliferation capacity. Altogether, our results suggest that the late sodium current due to $Na_v1.5$ QKP1510-1512 deletion

regulates ventricular fibroblasts proliferation. Finally, $Scn5a$ knockdown with ASO in ventricular fibroblasts from $Scn5a^{+/\Delta QKP}$ mice was associated with decreased $Tgf-β1$ expression ($P = 0.0167$; Fig. 4f). Furthermore, treatment with GW788388 plus ranolazine did not further decrease $Scn5a^{+/\Delta QKP}$ ventricular fibroblasts proliferation, suggesting associated signaling pathways (Supplementary Fig. 8)

## Increased proliferation of $Scn5a^{+/\Delta QKP}$ ventricular fibroblasts involves abnormal intracellular Ca$^{2+}$ handling

In cardiomyocytes, an increase in late sodium current can lead to an increase in the intracellular concentration of sodium promoting Na$^+$/Ca$^{2+}$ exchanger (NCX) to operate in reverse mode leading to an increase in intracellular calcium[20–22]. Previous study of $Na_v1.5$ in non-excitable cells has shown this mechanism to lead to a calcium increase that contributes to cellular proliferation[23]. Thus, we hypothesized that this mechanism could be involved in $Scn5a^{+/\Delta QKP}$ ventricular fibroblasts. When NCX was inhibited with 10 µM of YM244769, we observed a decrease of $Scn5a^{+/\Delta QKP}$ fibroblast proliferation similar to those observed with TTX or ranolazine ($P < 0.0001$; Fig. 5a). Dual inhibition of NCX (YM244769) and $Na_v1.5$ (ranolazine) did not exhibit cumulative effects on proliferation of $Scn5a^{+/\Delta QKP}$ ventricular fibroblasts (Supplementary Fig. 9a). In contrast, the YM244769 did not change control fibroblast proliferation (Supplementary Fig. 9b). To go further, we characterized the expression of NCX and different proteins involved in Ca$^{2+}$ signaling: calmodulin which is activated when forming a calcium/calmodulin complex, the calcium/calmodulin-dependent protein kinase II, and its phosphorylated form (P-CaMKII), and calcineurin[24–27]. $Scn5a^{+/\Delta QKP}$ cardiac fibroblasts exhibited a higher NCX expression than control fibroblasts ($P = 0.0306$) and no modification of calmodulin and calcineurin expression (Fig. 5b). They also exhibited a lower ratio of P-CaMKII to total CaMKII suggesting a downregulation of the CaMKII pathway (Fig. 5b), which could favor calcineurin activation. Therefore, we investigated the calcineurin pathway and showed that inhibiting calcineurin with 10 µM of iCaN induced a significant reduction of $Scn5a^{+/\Delta QKP}$ ventricular fibroblasts proliferation ($P < 0.0001$; Fig. 5a), while this inhibition did not impact control fibroblasts (Supplementary Fig. 9c). We also showed higher total NFAT protein levels in $Scn5a^{+/\Delta QKP}$ ventricular fibroblasts (Supplementary Fig. 9d). Finally, we investigated whether Ca$^{2+}$ homeostasis was altered. Activation of purinergic receptors with adenosine triphosphate (ATP) was used to induce Ca$^{2+}$ release from the endoplasmic reticulum via IP3 receptors, as classically performed[28–30]. As shown in Fig. 5c, $Scn5a^{+/\Delta QKP}$ ventricular fibroblasts had more frequent oscillations of intracellular Ca$^{2+}$ ($P = 0.0189$) and longer Ca$^{2+}$ transients than control fibroblasts ($P = 0.0418$).

## Discussion

Cardiac fibroblasts are abundant in heart[7] and play numerous roles in cardiac pathophysiology[31]. In the present study, we show that ventricular fibroblasts from the LQT3 mouse model exhibit, in culture, a higher proliferation rate and cell survival, and a lower differentiation into myofibroblasts, nevertheless associated with a higher expression of TGF-β and the activation of its canonical pathway leading to increased expression of collagens type I and III. In addition, this cell phenotype is, at least partly, linked to the expression of $Na_v1.5$ gain-of-function mutation in these non-excitable cells.

Cardiac fibroblasts express various ion channels[32–34]. For instance, they express K$^+$ channels which are key determinants of their resting membrane potential[35,36], and can modulate their proliferation and secretion[36,37]. It has also been shown that $Na_v1.5$ is also expressed in human atrial fibroblasts[11,12,14] and ventricular fibroblast cell line[13], although its role in these cells remained unknown. Here, we confirm the expression of $Scn5a$ in primary cultures of mouse ventricular fibroblasts at the gene and protein levels and show that the ΔQKP mutation of $Scn5a$ increases their proliferation and collagen secretion capacity. Although cardiac fibroblasts are not electrically excitable, their resting membrane potential varies from -20 mV to -50 mV depending on the studies[38,39]. These significant variations of

**Fig. 3 | Activation of transforming growth factor β (TGF-β) canonical pathway in *Scn5a*^+/ΔQKP ventricular fibroblasts. a** Representative western blots and protein expression, in fibroblasts from control and *Scn5a*^+/ΔQKP hearts, of TGF-β (*n* = 17 & 14, respectively), TGF- β receptor 1 (*n* = 21 & 13, respectively), Smad 2/3 and its phosphorylated form (P-Smad 2/3) and ratio; (*n* = 24 & 15, respectively), connective tissue growth factor (CTGF; *n* = 16 & 11, respectively), collagen I (Col I; *n* = 20 & 12, respectively), and collagen III (Col III; *n* = 19 & 17, respectively). **b** Representative western blots and protein expression in left ventricles from control and *Scn5a*^+/ΔQKP hearts, of TGF-β (*n* = 18 & 10, respectively), TGF- β receptor 1 (*n* = 13 & 8, respectively), P-Smad2/3 / Smad2/3 ratio (*n* = 11 & 12), CTGF (*n* = 11 & 12, respectively), Col I (*n* = 10 & 5, respectively), and Col III (*n* = 12 & 8). Data obtained from 4-week-old mice. The median and interquartile range are shown. Dots indicate values of each independent mouse. *, **, ***$p$ < 0.05, 0.01 and 0.001, respectively (Mann-Whitney test).

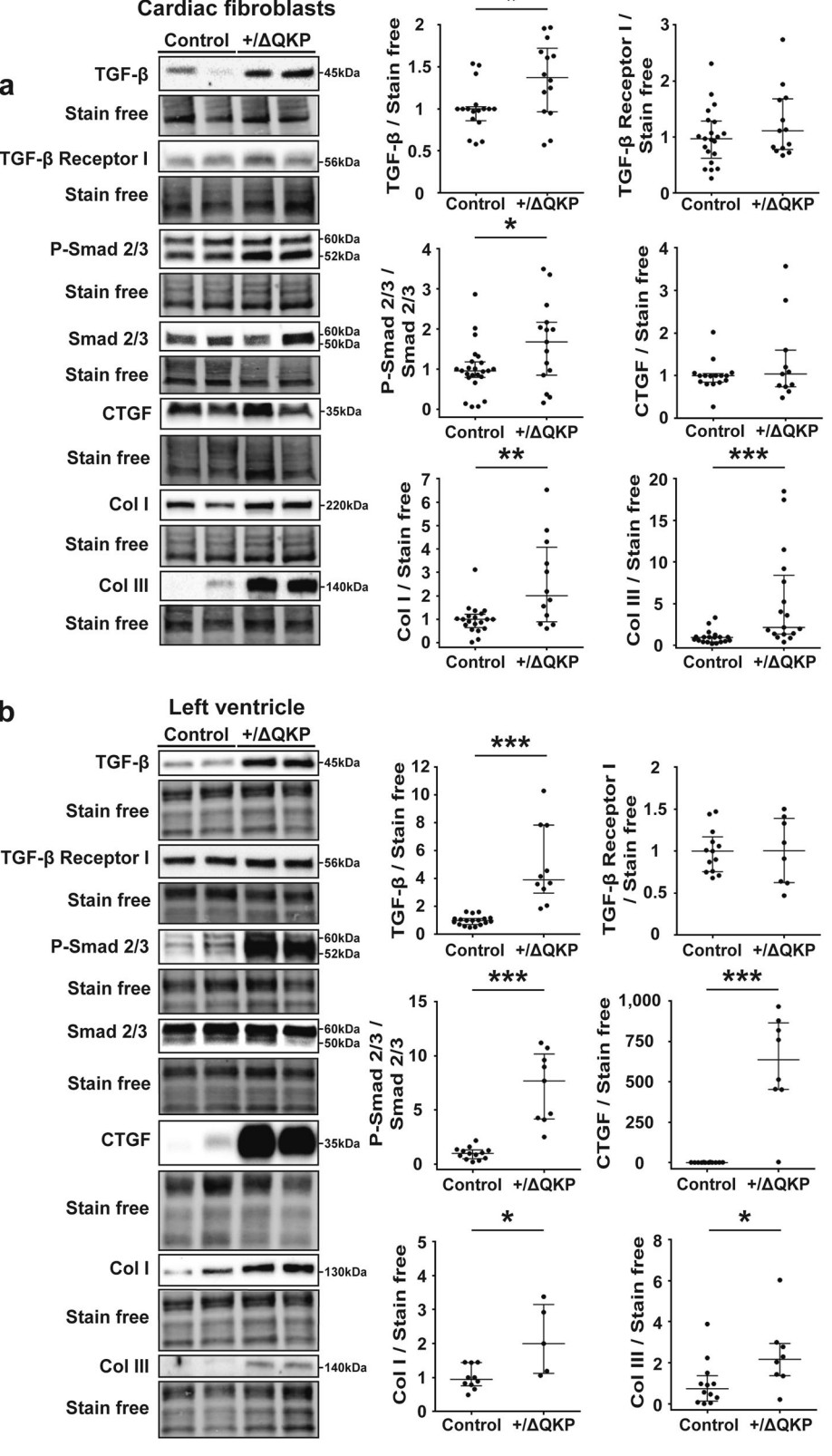

membrane potential can be explained by variable expression levels of ion channels depending on fibroblast differentiation state[36–40]. This voltage range corresponds to the sodium window current, which is increased by the ΔQKP mutation[5]. Moreover, the mutation induces a persistent current even at less negative membrane potentials.

Several studies have shown the expression of Na$_v$1.5 and the role played by the current it generates in non-excitable cells. For instance, Na$_v$1.5 activity participates to proliferation[41], survival[42], differentiation[23], or secretion[43] of different cell types. However, to the best of our knowledge, this is the first study showing a functional role of Na$_v$1.5 current in cardiac

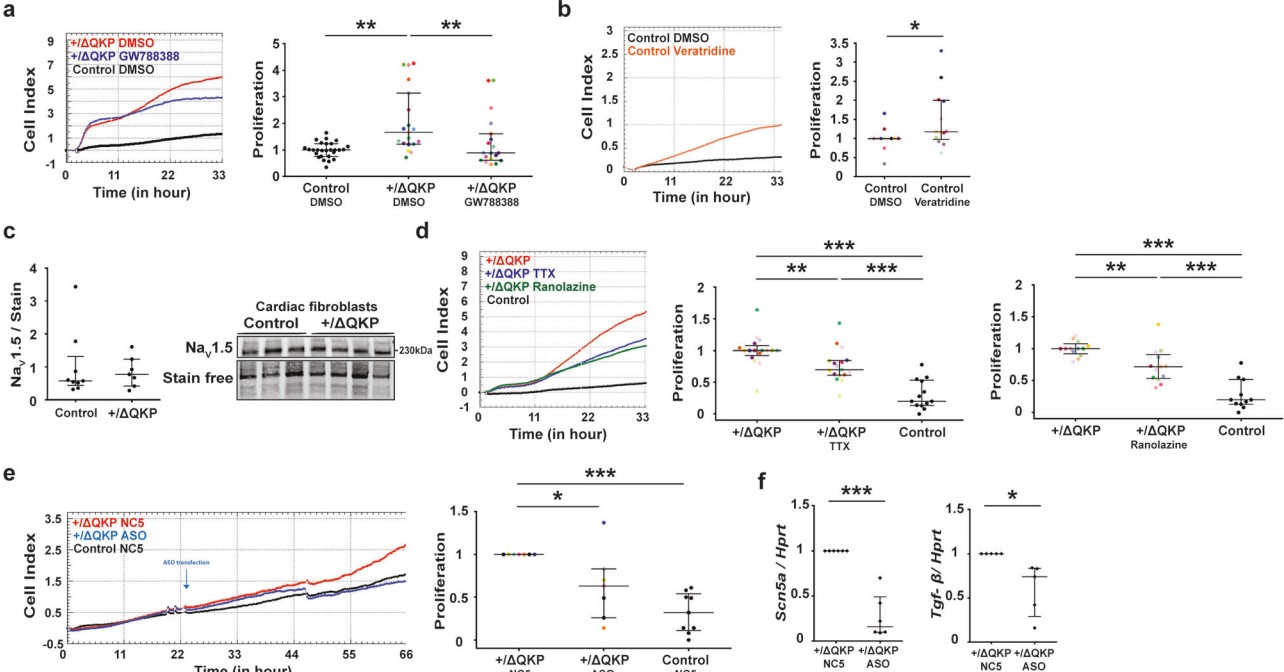

**Fig. 4 | Increased proliferative capacity of $Scn5a^{+/\Delta QKP}$ ventricular fibroblasts is related to $Scn5a$ gain of function.** Representative xCELLigence cell index time course and proliferation rate for (**a**) $Scn5a^{+/\Delta QKP}$ fibroblasts ($n = 19$ hearts), treated by 20 μM of GW788388 or vehicle (DMSO), normalized to mean value of control fibroblasts treated with DMSO ($n = 27$), (**b**) control fibroblasts ($n = 13$) treated with 1 μM of veratridine, or vehicle (DMSO; data normalized to control DMSO mean value). **c** Representative western blot and protein expression in cardiac fibroblasts from control and $Scn5a^{+/\Delta QKP}$ mice of $Na_v1.5$ ($n = 9$ & 7, respectively). **d** Representative xCELLigence cell index time course and proliferation rate for $Scn5a^{+/\Delta QKP}$ fibroblasts treated with 50 μM tetrodotoxin (TTX, $n = 17$), or 50 μM ranolazine ($n = 13$) and control fibroblasts ($n = 13$ & 11, respectively; data normalized to

$Scn5a^{+/\Delta QKP}$ mean value), and (**e**) $Scn5a^{+/\Delta QKP}$ fibroblasts transfected with the negative control NC5 ($n = 7$) or anti-$Scn5a$ ASO ($n = 7$), and NC5-transfected control fibroblasts ($n = 9$, each; data normalized to NC5-transfected $Scn5a^{+/\Delta QKP}$ mean value). Experiments performed with the same mouse are indicated by the same color. (**f**) $Scn5a$ ($n = 6$) and $Tgf-\beta$ ($n = 5$) mRNA expression in fibroblasts from $Scn5a^{+/\Delta QKP}$ ventricles transfected with the negative control NC5 or anti-$Scn5a$ ASO, relative to $Hprt$ and normalized to NC5-$Scn5a^{+/\Delta QKP}$ mean value ($2^{-\Delta\Delta Ct}$). Data obtained from 4-week-old mice. The median and interquartile range are shown. The dots indicate the values for each independent mouse. *, **, ***$p < 0.05$, 0.01 and 0.001, respectively (Kruskal-Wallis test in a-d & Mann-Whitney test in **e**).

fibroblasts. Indeed, our results show that inhibiting this current with TTX or ranolazine decreases the proliferation of $Scn5a^{+/\Delta QKP}$ mouse fibroblasts. In contrast, veratridine increases the proliferation of control fibroblasts. Veratridine has already been shown to induce a late sodium current in non-excitable cells and alter their functional properties. For instance, veratridine has been shown to increase the late sodium current and dose-dependently enhance the invasiveness of cancer cells, an effect antagonized by tetrodotoxin[44]. It has also been shown to induce membrane potential oscillations in non-excitable glioma cells, resulting in a depolarization of the membrane potential, which is inhibited by tetrodotoxin[45]. Thus, in vitro, activation of $Na_v1.5$ and changes in membrane potentials could be induced by veratridine. Altogether, these results show the implication of the late sodium current in cardiac fibroblast proliferation. However, the absence of the effect of TTX or ranolazine on control fibroblasts suggests that the sodium current does not significantly contribute to basal proliferation.

Our study also shows that the effect of the late sodium current on proliferation involves TGF-β pathway. Indeed, on one hand, TGF-β expression is decreased when $Scn5a$ expression is inhibited, and, on the other hand, inhibition of TGF-β receptor prevents fibroblast proliferation. Moreover, when $Scn5a^{+/\Delta QKP}$ fibroblasts were treated with both ranolazine and GW788388, no cumulative effect was observed showing that these two actors are connected. Kaur and colleagues showed that cardiomyocyte treatment with TGF-β1 increased the $Scn5a$ transcription and sodium current via PI3K/Akt-mediated phosphorylation of foxO[46]. Here, we show that, conversely, the increase in sodium current also activates TGF-β expression in $Scn5a^{+/\Delta QKP}$ fibroblasts.

Any condition leading to increased intracellular sodium concentration may enhance the reverse mode activity of NCX and consequently results in

increased intracellular calcium concentration[47–49]. As a consequence, different pathways such as protein phosphatase calcineurin signaling can be activated by $Ca^{2+}$ and alter gene transcription programs, including those involved in fibrosis development, via the nuclear factor of activated T cells (NFAT)[50–52]. Our results show that these mechanisms are involved in the activation and proliferation of $Scn5a^{+/\Delta QKP}$ ventricular fibroblasts. However, our results also suggest that $Na_v1.5$ late current cannot fully explain on its own the increase of proliferation of $Scn5a^{+/\Delta QKP}$ fibroblasts since the inhibition of $Na_v1.5$ prevents it only partly while the inhibition of TGF-β receptor prevents it fully, suggesting that another mechanism is involved in TGF-β higher expression in parallel to $Na_v1.5$ effect.

Cardiac fibroblasts regulate the homeostasis of the extracellular matrix and cell-to-cell communication[8,53,54]. They also modulate cardiomyocyte structure and electrical activity by direct interactions and/or paracrine factors[9,10,55–57]. Here, we observed higher proliferation and survival rates of ventricular fibroblasts from $Scn5a^{+/\Delta QKP}$ mice not only in vitro, but also in vivo, in $Scn5a^{+/\Delta QKP}$ mice ventricles. In addition, our in-vitro experiments showed higher TGF-β production by $Scn5a^{+/\Delta QKP}$ fibroblasts associated with the activation of its canonical pathway and collagen I and III production. These results, confirmed in vivo, are consistent with the development of ventricular fibrosis that we observed in $Scn5a^{+/\Delta QKP}$ mice[5].

TGF-β is known to act on fibroblasts, inducing changes of proliferation, differentiation, secretion, and membrane potential[15,58–60]. Here, we showed the autocrine role of TGF-β to promote fibroblast proliferation, prevented by TGF-β receptor inhibition, in $Scn5a^{+/\Delta QKP}$ fibroblasts. The size of the fibroblast population contributes to the heart size, especially during physiological cardiac development[61]. So, the increase of heart size observed in $Scn5a^{+/\Delta QKP}$ mice might be explained not only by cardiomyocyte

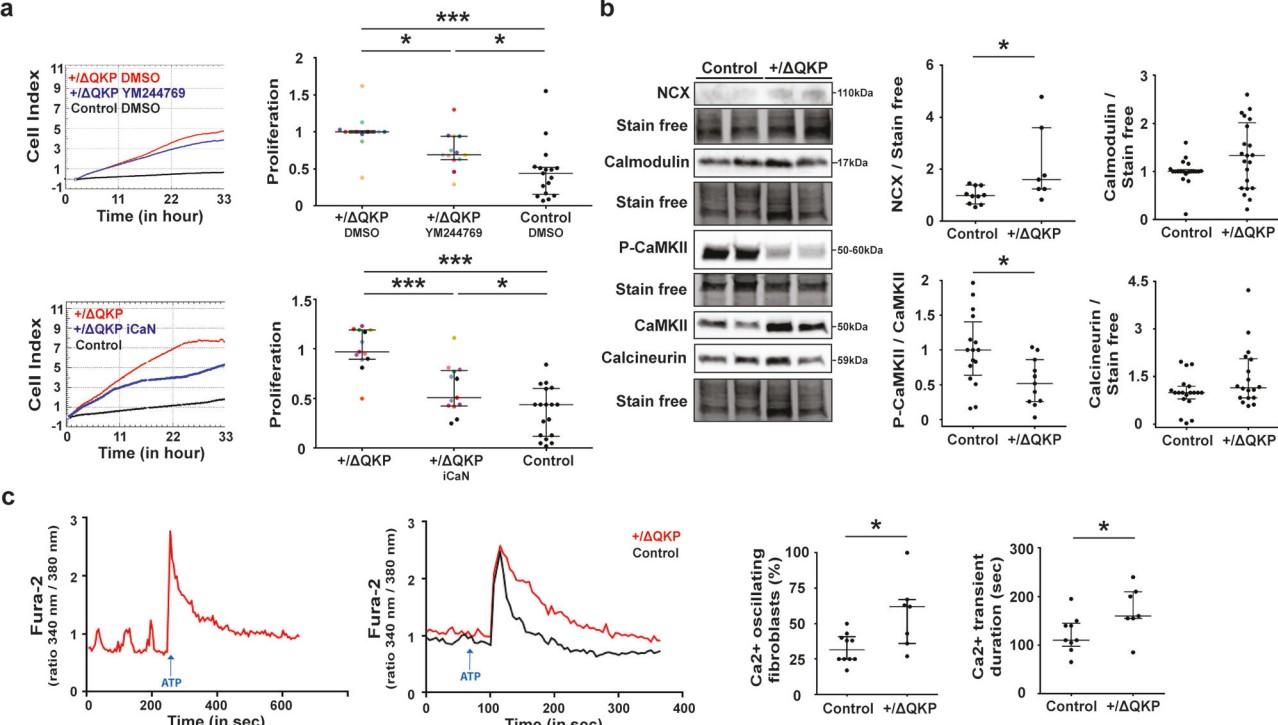

**Fig. 5 | Larger proliferative capacity of ventricular fibroblasts from $Scn5a^{+/\Delta QKP}$ mice is related to altered calcium homeostasis. a** Representative xCELLigence cell index time course and proliferation rate for $Scn5a^{+/\Delta QKP}$ fibroblasts treated with 10 µM YM244769 or vehicle (DMSO; n = 13 & 17, respectively) and control fibroblasts (DMSO; n = 12); and for $Scn5a^{+/\Delta QKP}$ fibroblasts treated with 10 µM iCaN or not (n = 13 & 17, respectively) and control fibroblasts (n = 18). Data are normalized to DMSO- or untreated $Scn5a^{+/\Delta QKP}$ mean value. Experiments performed with the same mouse are indicated by the same color. **b** Representative western blots and protein expression in fibroblasts from control and $Scn5a^{+/\Delta QKP}$ hearts of NCX (sodium-calcium exchanger; n = 10 & 7, respectively), calmodulin (n = 20 & 21,

respectively), phosphorylated-CamKII/CaMKII ratio (Calcium/calmodulin-dependent protein kinase II; n = 16 & 10, respectively), and calcineurin (n = 19 & 18, respectively). **c** Representative time course of intracellular $Ca^{2+}$ concentration ($[Ca^{2+}]_i$) at baseline and after ATP stimulation ($Ca^{2+}$ transient; top). Percentage of fibroblasts with diastolic $[Ca^{2+}]_i$ oscillations, and duration of ATP-induced $Ca^{2+}$ transient from control and $Scn5a^{+/\Delta QKP}$ hearts (n = 10 & 7, respectively). Data obtained from 4-week-old mice. The median and interquartile range are shown. The dots indicate the values for each independent mouse. *, ***$p < 0.05$ and 0.001, respectively (Kruskal-Wallis test in **a**, and Mann-Whitney test in **b**, **c**).

hypertrophy and ventricular dilation, as previously shown by our team[5], but also by a larger population of fibroblasts.

A larger proportion of fibroblasts within the heart tissue can also have functional consequences. Indeed, fibroblasts, as non-excitable and non-conductive cells, can electrically isolate cardiomyocytes[62,63]. They can also depolarize the membrane of cardiomyocytes and prolong their action potential[53]. In addition, cardiac fibroblast paracrine factors have been shown to alter cardiomyocyte channel expression and excitability[9]. More particularly, TGF-β can decrease HERG and Kir2.1 expression[6] and increase Na$_v$1.5 expression in cardiomyocytes[46], in addition to promoting cardiomyocytes hypertrophy[56,64,65]. All these modifications may further prolong the cardiomyocytes action potential in addition to the effects of the late Na$^+$ current due to the LQT3 Na$_v$1.5 mutation[5] and contribute to occurrence of arrhythmias[66–69]. Furthermore, the higher expression of *Il-1β* and *Il-6* in $Scn5a^{+/\Delta QKP}$ ventricular fibroblasts may also contribute to the pathological process by promoting cardiac fibroblast activation[70], altering calcium signaling[71] and favoring cardiomyocytes hypertrophy[72]. Cx43 lower expression in $Scn5a^{+/\Delta QKP}$ mice also contributes to rhythm disorders by altering the cardiomyocyte electrical coupling[73] and promoting fibrosis as described in other cardiac diseases[74,75].

We also observed a significant increase in CTGF expression in $Scn5a^{+/\Delta QKP}$ ventricles but not in fibroblast culture. This growth factor often acts in combination with TGF-β during fibrotic remodeling[17]. This highlights the involvement of other cell types such as cardiomyocytes or endothelial cells, in these pathophysiological processes[76,77]. This increase in CTGF could contribute to the activation of fibroblasts within the tissue and also to cardiomyocyte hypertrophy[78–80].

The proliferation and production of profibrotic factors are characteristic of activated fibroblasts, *i.e.*, myofibroblasts. However, we observed a lower α-SMA level, a marker of myofibroblasts, in $Scn5a^{+/\Delta QKP}$ fibroblasts in vitro, suggesting a lower ability to differentiate into myofibroblasts. This result was confirmed in vivo. At first glance, this might appear paradoxical. However, Liu and collaborators similarly observed active non-contractile pulmonary fibroblasts leading to fibrosis, despite an absence of α-SMA expression[81]. Our hypothesis is that the $Scn5a^{+/\Delta QKP}$ fibroblasts could be at a proto-myofibroblast stage. Indeed, during their differentiation, fibroblasts undergo an intermediate stage of proto-myofibroblasts which proliferate, migrate, and increase their production of ECM and cytokines before fully differentiating into myofibroblasts. This transitional and activated state is still poorly characterized[82]. A known feature of proto-myofibroblasts is a rearrangement of the actin cytoskeleton with the formation of stress fibers that extend along the length of these cells. The absence of α-SMA within these stress fibers distinguishes proto-myofibroblasts from myofibroblasts which do exhibit α-SMA-positive stress fibers. Moreover, myofibroblasts, in contrast to proto-myofibroblasts, are characterized by low migration and proliferation capacities, and increased contractility[83,84]. The increased proliferation rate, ECM production, and formation of stress fibers, with a low expression of α-SMA, in $Scn5a^{+/\Delta QKP}$ fibroblasts support our hypothesis that they are proto-myofibroblasts[15,85,86]. Whatever the case, it is widely acknowledged that the majority of cardiac fibroblasts form via Epithelial-to-Mesenchymal Transition (EMT)[87], and Na$_v$1.5 has been shown to regulate EMT and invasiveness in breast cancer cells[88]. Further studies are warranted to ascertain whether Na$_v$1.5 is also involved in the transition

between fibroblasts, proto-myofibroblasts, and myofibroblasts, as suggested by our findings.

In conclusion, our study shows modifications of cardiac fibroblast phenotype in LQT3 which lead to cardiac structural anomalies associated with an increase in fibroblast proliferation. In addition, fibrosis is secondary to the production of TGF-β, the activation of its canonical pathway, and the production of collagens I and III by fibroblasts. This could contribute to the increased risk of arrhythmias in LQT. We also demonstrate the expression of *Scn5a* in mouse ventricular fibroblasts, and, to the best of our knowledge, show for the first time that a Na$_v$1.5 gain-of-function mutation, leading to a larger late sodium current, can alter fibroblast proliferation.

## Methods

### Animals

The $Scn5a^{+/\Delta QKP}$ mouse model was generated in PolyGene AG facilities as previously described[5]. Animal experiments were performed in the animal facility of Nantes University Health Research Institute (UTE – IRS-UN) which has been accredited by the French Ministry of Agriculture. The experimental procedures were approved by the regional ethics committee (CEEA – Pays de la Loire, France) according to the Directive 2010/63/EU of the European Union. We have complied with all relevant ethical regulations for animal use.

All the experiments were performed on $Scn5a^{+/\Delta QKP}$ mice and on $Scn5a^{+/+}$ (wildtype) and $Scn5a^{+/+}$ -Flp mice as controls. All mice were on a C57Bl6/J genetic background. Both male and female mice have been utilized in this study. The $Scn5a^{+/+}$ -Flp mice have a wildtype *Scn5a* locus but express the flipase gene, which is also expressed in $Scn5a^{+/\Delta QKP}$ mice. Previously, we showed no difference in cardiac phenotype between $Scn5a^{+/+}$ and $Scn5a^{+/+}$ -Flp mice[5]. In this study, we observed no difference between ventricular fibroblasts from $Scn5a^{+/+}$ and $Scn5a^{+/+}$-Flp mice (Online Supplementary Figs. 10-13). Therefore, we pulled the data from the two groups in the same control group.

### Isolation and culture of ventricular fibroblast

Ventricular fibroblasts were isolated from 4-week-old mice by using the classical Langendorff technique. Ten minutes after being heparinized (100 μL heparin Panpharma, 5000 UI/mL), mice were sacrificed and their heart quickly excised and dropped in a cold solution containing (in mM): NaCl, 120; KCl, 5.4; MgSO$_4$, 2.45; NaH$_2$PO$_4$, 1.2; HEPES, 1.2; Glucose, 5.6; 2,3-Butanedione 2-monoxime (BDM), 10; Taurine, 5; pH 7.4 with NaOH. After cannulation of the aorta, the heart was perfused for 1 min on a Langendorff system (37 °C) with the same solution supplemented with 1 mM of CaCl$_2$, followed by 5 minutes of perfusion with a solution without CaCl$_2$. Then, the heart was perfused with a low-CaCl$_2$ solution (12.5 μM) containing 28 U of type 14 protease (Sigma® P5147) and 8500 U of type 2 collagenase (Worthington® LS00477) for 12-16 minutes. The digested heart was cut below the atria, and the ventricles were recovered in the digestion solution and gently triturated. Then a "stop" solution (perfusion solution with 10% of Fetal Calf Serum (FCS) and 12.5 μM of CaCl$_2$) was added and cardiomyocytes let settled down during 10 minutes. Then the supernatant with cardiac fibroblasts was collected in a 50-mL conical tube and centrifuged at $500 \times g$ for 10 minutes at room temperature. After centrifugation, the supernatant was collected and re-centrifuged at 1000 x $g$ for 10 minutes. The two cell pellets were re-suspended together in 50 mL of DMEM medium (Dulbecco's Modified Eagle Medium with 4.5 g/L of D-Glucose, 4 mM of L-Glutamine, and 1 mM of Pyruvate; Gibco® #41966-094) supplemented with 10% of FCS, 1% of antibiotics (Penicillin, Streptomycin) and 2 mM of glutamin. Then 10 mL of the cell suspension were seeded in 100-mm plastic cell culture dishes (Nunc®, P7741) and incubated at 37 °C with 5% CO$_2$. After 24 hours, the cell cultures were washed with 1X phosphate-buffered saline (PBS Gibco® #14190-094) and fresh supplemented DMEM medium was added. Non-adherent cardiomyocytes were removed by changing the culture medium. Every other day, the cells were rinsed once with PBS and a fresh medium was added.

The quality of the fibroblast primary culture was controlled at days 5 and 8 of culture with immunofluorescent detection of VE-cadherin (marker of endothelial cells) and staining of F-actin with rhodamin-phalloidin, showing a large majority of fibroblasts at days 5 and 8 of culture (Online Supplementary Fig. 13).

### Impedance-based xCELLigence proliferation assay

The xCELLigence system (impedance-based real-time cell analyzer - RTCA Roche, ACEA Biosciences Inc.™) was used to study in real-time ventricular fibroblast proliferation[89], according to the manufacturer's instructions. On day 5 of the culture, fibroblasts were washed with 1X PBS (without calcium and magnesium) and isolated with 2.5% of trypsin-EDTA during 5 minutes at 37 °C. One mL of culture medium was added to stop the trypsin action and collect the fibroblasts. Fibroblasts were quantified with a Malassez counting chamber. For xCELLigence proliferation assay, the background impedance of culture medium in an E-plate View (ACEA Biosciences Inc.™ – 16 or 96 wells) was measured with 100 μL of culture medium per well. Then, fibroblasts were seeded into the E-plate at a density of 5000 cells/well. The plate was set on the RTCA station, in a humidified incubator at 37 °C (5% CO$_2$). Proliferation was monitored every 15 min for up to 70 h by the RTCA Analyzer, and the culture medium was changed every day during the experiment. Data were analyzed with the RTCA Software 1.2.1. and proliferation rate was measured between 20 and 25 h after seeding. Relative proliferation rates were normalized as mentioned in legends.

### Veratridine, tetrodotoxin, ranolazine, GW788388, YM244769 and Calcineurin autoinhibitory peptide treatments

Fibroblasts were chronically treated with veratridine (1 μM) diluted in DMSO (final concentration of 0.001% - Alomone) from the beginning of cell culture (5 days before seeding in e-plate), or with tetrodotoxin (50 μM - TTX citrate - Tocris Bioscience), ranolazine (50 μM - Tocris Bioscience) or calcineurin autoinhibitory peptide diluted in media (iCaN - Tocris Bioscience - 10 μM), GW788388 (20 μM) diluted in DMSO (final concentration of 0.00003%), or YM244769 (Tocris Bioscience - 10 μM) diluted in DMSO (final concentration of 0.01%) from the beginning of the xCELLigence experiment (day 5 of culture). Untreated fibroblasts or treated with DMSO were used as reference. Each experiment was performed in duplicate.

### Antisense oligonucleotide transfection

Antisense Oligonucleotide against *Scn5a* (ASO; Sequence: GACTATGC-CATCGGCCTCCA) and control scrambled oligonucleotide (NC5; Sequence: GCGACTATACGCGCAATATG) were designed and produced by Integrated DNA Technologies (Belgium). The transfection was realized at day 6 of culture on fibroblasts in classical culture. For xCELLigence assays, fibroblasts were plated in E-plate View on day 5, and transfected 24 hours after (day 6). One hour before transfection, the medium was changed with fresh DMEM medium for fibroblasts. Oligonucleotides were diluted at 10 nM in fibroblast medium without serum and mixed with HiPerfect (Qiagen) (8 μL ASO at 10 μM, 6 μL HiPerfect, 200 μL DMEM without serum). After 10-min incubation at room temperature, the oligonucleotide/transfection agent solution was added to the cells and incubated for 24 h, before being washed with fresh medium. Experiments were performed 24 hours later, on day 8.

### Measurement of intracellular free [Ca$^{2+}$]$_i$

Fibroblasts were harvested at day 5 of culture, then seeded in u-Slide 8 Well ibidi plates (80826 - Ibidi) at a density of 10,000 cells/well. Fifteen hours later, cells were washed with 1X HBSS (5 mM HEPES, 2% SVF, +1.5 g/L Glucose, 1.26 mM CaCl$_2$), and incubated in the same medium with 5 μM Fura-2 AM for 1 h at 37 °C in the dark. The cells were washed and incubated with the same medium as before. Fura-2 was excited at 340 nm and 380 nm, and the emission was detected at 510 nm. Adenosine triphosphate (ATP) was used as a traditional chemical stimulation to study calcium signaling in fibroblasts by inducing a calcium transient[28–30]. Briefly, 100 μM ATP was

applied to induce an increase in $[Ca^{2+}]_i$, measured at 37 °C. $[Ca^{2+}]_i$, expressed as 340/380 nm excitation ratio was monitored using a widefield microscope (Leica DMI 6000B) and MetaFluor software.

## RNA extraction and reverse transcriptase-polymerase chain reaction (RT-PCR)

Total RNA was extracted from ventricular fibroblasts using Nucleospin RNA Kit (Macherey-Nagel - #740955.50) at day 8 of culture, and treated with RNase-free DNase as *per* the manufacturer's instructions. Reverse transcription was performed using SuperScript™ VILO™ cDNA synthesis Kit (Invitrogen™ – #11756050). Then, PCR was performed with the 7900HT Fats Real-Time PCR System (Applied Biosystems) using TaqMan® universal PCR Master Mix (Applied Biosystems – #4304437), with the following thermal cycling conditions: 2 min at 50 °C, 10 min at 95 °C, followed by 40 cycles at 95 °C for 15 sec, 60 °C for 1 min. TaqMan® probes to mouse *Acta2* (assay ID: #Mm01546133), *Gja1* (#Mm00439105), *Il-1α* (#Mm00439620), *Il-1β* (#Mm00434228), *Il-6* (#Mm00446190), *Scn5a* (#Mm01342518) and *Tgf-β1* (#Mm01178820) were used to amplify and detect the corresponding genes. Quantification of each mRNA was carried out with *Hprt* (#Mm01545399) as the reference gene, using the $\Delta\Delta C_T$ method.

## Western Blot

Whole cell lysates were obtained using 1% Triton X-100 buffer (in mmol/L: NaCl, 100; Tris-HCl, 50; EGTA, 1; $Na_3VO_4$, 1; NaF, 50; phenylmethylsulfonyl fluoride, 1 (Roche Applied Science); and protease inhibitor mixture (Sigma P8640; 1:100); pH 7.4.) on ice. Samples were sonicated, and centrifuged at $15,000 \times g$ for 15 min at 4 °C. Total protein was quantified using the Pierce BCA Protein Assay Kit (Thermo Scientific, Belgium). Equal samples were loaded and separated by electrophoresis. Samples were prepared with 35 µg of total protein, 1X of NuPAGE Sample Reducing Agent (Invitrogen), 1X of NuPAGE LDS Sample Buffer (Invitrogen) and boiled for 5 min. Samples were run on 4-15% Mini-PROTEAN® TGX Stain-Free™ Precast Gels (Bio-rad) and transferred on Trans-Blot® Turbo™ Nitrocellulose Transfer Packs (Bio-rad). Membranes were blocked (using 5% non-fat milk) and incubated with primary antibodies targeted against Vimentin (Cell Signaling, #5741; 1: 1000), Cleaved caspase 3 (Cell Signaling, #9664; 1: 250), Caspase 3 (Cell Signaling, #9662; 1: 250), Mcl-1 (Cell Signaling, #5453BC; 1: 500), α-SMA (Sigma-Aldrich, #A5228; 1: 1000), Periostin (Abcam, #ab14041; 1: 1000), Connexin 43 (Sigma Aldrich, #C6219; 1: 10000), TGF-β (Cell Signaling, #3711S; 1: 500), TGF- β receptor 1 (Abcam, #ab31013; 1: 500), CTGF (Santa Cruz, #1439; 1: 500), Smad-2/3 (Millipore, #07-408; 1: 250), P-Smad-2/3 (Cell Signaling, #8828S; 1: 250), Collagen Type I (Cell Signaling, #84336; 1: 500), or Collagen Type III (Rockland, #600-401-105; 1: 500). Then, membranes were incubated with the ad-hoc secondary horseradish peroxidase antibody (Anti-Goat Santa Cruz, #sc-2922, Anti-Mouse Cell signaling, #7076, and Anti-Rabbit Cell signaling, #7074; 1: 5000). Incubation was followed by detection using chemiluminescence (Clarity™ Western ECL Substrate, ChemiDoc™ MP System Bio-rad). Quantification was performed on unsaturated images with Image Lab™ Software (Bio-rad). The relative expression of the protein of interest to the total protein amount (stain-free method) in the corresponding lane was normalized to the $Scn5a^{+/+}$ or control mean value. Uncropped and unmodified blot images are available in the Supplementary Information.

## Immunohistochemistry

Immunostaining was performed on cultured ventricular fibroblasts in Ibidi plates (15 µ-Slide-well – 80826, Ibidi GMBH) at day 8 of culture. Rhodamin-phalloidin staining (Invitrogen, #R415, 1: 1000) and immunostaining of α-SMA (Sigma-Aldrich, #A5228; 1: 1000) were used to appreciate the stage of differentiation into myofibroblasts and VE-cadherin immunostaining (Santa Cruz, #sc6458, 1: 500) to evaluate the endothelial population in the fibroblast culture. Cells were fixed 10 minutes in 4% paraformaldehyde at room temperature and after being washed with phosphate-buffered saline 1X (PBS1X), they were permeabilized and blocked (BSA 1%, triton 0.3%,

fish gelatin 3%, PBS1X) during 1 hour at room temperature, and subsequently incubated overnight at 4 °C in a humidified chamber with primary antibodies. Then, after being washed with PBS1X, cells were incubated with fluorochrome-conjugated secondary antibodies (Alexa Fluor 488 Goat anti-Mouse #A11001, Alexa Fluor 568 Goat anti-Rabbit #A11011, Alexa Fluor 488 Donkey anti-Goat #A11055; 1: 1000) and phalloidin at room temperature for 1 hour, followed by nuclei staining with Hoechst (Sigma-Alrdich #B2883, 1: 1000) at room temperature for 10 minutes. Wells were washed and loaded with 0.1% paraformaldehyde to preserve fluorescence. Imaging was performed on an Axiovert 200 M Zeiss microscope and captured with Axiovision Rel 4.5 software. Image analysis was performed using ImageJ software 1.45b (NIH Software).

For heart tissue analyses, 4-week-old mice were euthanized and hearts were isolated and rinsed in cold phosphate buffered saline ($Na_3VO_4$ 0.5 mM, NaF 5 mM, PBS1X without calcium and magnesium). The hearts were dried, and then quickly frozen in isopentane cooled in liquid nitrogen. Transverse sections of 7 µm were performed with Cryostat (MICROM HM 560). The sections were transferred to glass slides. Essentially the same staining protocol was applied for cultured cells and cryosections, except that washes were performed with 0.3% Triton X-100 and slices were mounted with ProLong® Gold (P36934, Life Technologies). The heart cryosections were immunostained with vimentin (Cell Signaling, #5741, 1: 250) and α-actinin (Sigma-Aldrich, #A7811, 1: 250) antibodies to visualize cardiac fibroblasts and cardiomyocytes, respectively, and with Ki67 (Millipore, #AB9260, 1: 250) a marker of proliferation. For each heart, 2 images were obtained in the left ventricular region, avoiding areas with vessels.

## Statistics and reproducibility

Data are expressed as median and interquartile range. The sample size expressed as n value provided in the manuscript corresponds to the number of mice used, and is indicated in the legends of each figure. At least 2–3 replicates were used per biological experiment. Statistical analysis was performed with Prism6 (GraphPad Software, Inc., USA). Statistical significance was determined with the Mann–Whitney rank sum test for the comparison of the two groups. For more than two groups, one-way ANOVA or Kruskal-Wallis test was performed with Bonferroni or Dunn post-hoc test when appropriate. A value of $p \leq 0.05$ was considered significant.

## Reporting summary

Further information on research design is available in the Nature Portfolio Reporting Summary linked to this article.

## Data availability

The source data used to generate the graphs in the main figures are provided as Supplementary Data. Uncropped blot images of this study are provided in the Supplementary Information. The data from this study are available from the corresponding author upon reasonable request.

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

## Acknowledgements

The authors thank Drs. Vincent Sauzeau and Florian Dilasser for scientific advices and the staff from the animal facility (UTE IRS-UN) for technical assistance. We also acknowledge the IBISA MicroPICell facility (Biogenouest), a member of the national infrastructure France-Bioimaging supported by the French national research agency (ANR-10-INBS-04). The

research leading to these results has received funding from the *Agence Nationale de la Recherche* (ANR-12-BSV1-0013-01 and ANR-19-CE14-0031-02 to FC), the DHU2020 (MD) and Genavie Foundation (MD).

## Author contributions

F.I.C. and M.D. conceived the research, obtained the funding, and assessed the results. C.C., J.P., C.O.C., A.T., E.L., C.J., Fr.C., and I.F. performed the experiments and analyzed the data. C.C., F.I.C., M.D., and I.B. wrote the manuscript. All authors edited the manuscript.

## Competing interests

The authors declare no competing interests.
