## [Transparent Peer Review file · Communications Biology]

Long QT syndrome type 3 gain-of-function of Nav1.5 increases ventricular fibroblasts proliferation and pro-fibrotic factors

Corresponding Author: Dr Mickael Derangeon

Version 0:

Reviewer comments:

Reviewer #1

(Remarks to the Author)

In the paper titled "Long QT syndrome type 3 gain-of-function of Nav1.5 increases ventricular fibroblasts proliferation and pro-fibrotic factors" by Castro et al, based on their previous study and combined with the clue of expression of SCN5A gene in cardiac fibroblasts, the author proposes a scientific question "the potential role of gain of function of Nav1.5 in cardiac fibroblasts". The results show that cardiac fibroblasts from Scn5a+/ΔQKP knock-in mice have stronger proliferation ability compared to the control mice, and this high proliferation ability is related to the activation of TGF signaling pathway. In addition, abnormal calcium signal activity could partially explain the high proliferation activity of cardiac fibroblasts from the Scn5a+/ΔQKP knock-in mice. This is an interesting question to be addressed because the role of this gene in the cardiac fibroblasts is still unknown. In this sense this paper adds a contribution to these findings. Before considering acceptance, some experiments need to be supplemented and the following concerns need to be addressed.

Major concerns:

1. The author used 4-week-old mice in the experiment. In addition to observation of obvious proliferation of cardiac fibroblasts in the +/ΔQKP group, there was also a significant phenotype of cardiac enlargement. It is necessary to provide experimental evidence to prove whether this enlargement is caused by an increased number of fibroblasts or hypertrophy of myocardial cells. Considering the important role of fibroblasts in myocardial hypertrophy, it is necessary to conduct the same experiment at more time points (such as Fig1), such as 1 week before birth, 1 and 2 weeks after birth. The dynamic data presentation is of great significance for readers to obtain more information and clues.
2. Although the author mentioned some seemingly contradictory aspects regarding fibroblast activation in the discussion and proposed a hypothesis that the fibroblasts in this study stimulated by gain of function of Nav1.5 may be at the stage of proto-fibroblasts, there are indeed some deviations from consensus in the results presented. For example: Scn5a+/ΔQKP mice have high levels of collagen secretion, high protein levels of CTCF and inflammatory factors in the heart, while α-SMA expression is down-regulated. It is strongly recommended that further and deeper discussion was added in the discussion section to discuss this point. In addition, some more experiments should be designed to prove the possible stage of proto-fibroblast, such as detecting specific proto-fibroblast-specific markers to preliminarily prove the hypothesis.
3. It is inferred that homozygous mice with ΔQKP may be embryonic lethal. It will be quite interesting to investigate the effect of conditional expression of ΔQKP in the fibroblast-specific or inducible manner on the phenotype of mice.
4. The results of immunofluorescence staining of cardiac myocytes and fibroblasts in Fig-S8 and Fig1 need to be improved, with inconsistent sections of representative regions in which inconsistent orientation of cardiac myocytes (transverse vs longitudinal). It is recommended to perform whole-heart sections on the coronal section (it will display the four chambers of the heart), based on which the results are more comprehensive, objective, and accurate.

Minor concerns:

1. In Fig2B, IF detection of cell proliferation should also include markers for cardiac fibroblasts to prove which Ki67-positive cells belong to cardiac fibroblasts but not the other cell types. Similarly, whole-heart sections should be performed on the coronal section.
2. In Fig2C, in addition to Western Blot, detection of apoptosis should also include whole-heart sections on the coronal

section using TUNNE or IF of cleaved caspase 3.

3. The author detect the protein level of SMAD2/3 using the method of Western Blot, and the result showed only one band. However, as the different family numbers of SMAD family, the result of Western Blot for SMAD2/3 should have two bands corresponding to SMAD2 and SMAD3. Please confirm the information of antibody used and address this issue.

4. In Fig2C, in addition to Western Blot, detection of apoptosis should also include whole-heart sections on the coronal section using TUNNE or IF of cleaved caspase 3.

Reviewer #2

(Remarks to the Author)

In this manuscript the authors extend their studies directed toward determining functional effects of sodium channel expression/activation in a cardiac fibroblast population isolated from the ventricles of adult mice engineered to express a mutant sodium channel that has been shown to have an enhanced late sodium current. The manuscript reports the phenotype of this fibroblast population in terms of markers for fibroblasts vs. myofibroblasts; the presence of TGF-beta receptors and its downstream signaling pathway; and some aspects of intracellular calcium signaling in these cells. After reading and then rereading this complex paper my assessment is that it presents a number of related and novel data sets and is directed toward an important translational pathophysiology. Unfortunately however, few if any of its main claims or conclusions are complete or convincing as presented. In revising and extending this work the authors will need to consider:

1. Whether their study can or should be described in terms of a sodium channel mutation with a sole emphasis on the alpha subunit of the Nav1.5 channel; or whether an extended study could or should also include analyses of the sodium channel complex - including the beta subunit composition in this study.

2. Adding information that would allow the reader/reviewer to have some idea of the sodium channel density in the isolated ventricular fibroblast preparation. In the absence of this it is not possible to relate the data sets generated by the indirect assays either to existing previously published data or to the functional ventricular substrate at baseline or in disease states.

3. Including more information concerning the actions of veratridine, including its concentration-dependent effects. It is unclear whether veratridine would be expected to enhance sodium current in a quiescent preparation such as the cardiac fibroblast, or even in a fibroblast preparation whose resting potential oscillates as described on page 7. In my experience, isolated cardiac fibroblasts in primary culture have a stable resting potential.

4. The section of the manuscript dealing with fibroblast calcium homeostasis is puzzling. Why was ATP at 100 micro molar chosen as the agonist and how does the putative activation of a number of different purinergic receptors relate to the main theme of this work?

5. The new or revised manuscript needs to be edited and clarified extensively. There are a number of key sentences or sections that are ambiguous as a result of the use of unconventional Scientific English. Examples include:

- sentences in the Abstract on lines 8 and 9 and lines 12 and 13
- the last two sentences of paragraph 1 in the Introduction on page 3
- the entire last paragraph of the Introduction on page 4
- the text on lines 9 through 13 on page 9
- the section describing TGF-beta results that is on page 10 between lines 12 and 18
- the use of the word 'fibroblasts' throughout the manuscript. This should be the singular 'fibroblast' in almost all cases.

Version 1:

Reviewer comments:

Reviewer #1

(Remarks to the Author)

The authors answered all my questions. I am satisfied with the revised manuscript.

Reviewer #2

(Remarks to the Author)

Thank you for considering my comments and in most cases making wording changes in the manuscript that partially address the point that my review raised.

Response to reviewers' comments

We thank the reviewers for their thorough evaluation of our manuscript and interesting comments and questions. We apologize for the time needed to answer but some comments required that we revived the Scn5a+/ Δ QKP knock-in mouse model and performed new experiments. The reactivation of the Scn5a+/ Δ QKP knock-in mouse model unfortunately took a considerable amount of time.

To facilitate the visualization of the changes made to the manuscript, we have highlighted in yellow the text added or modified following the reviewers' requests. Stylistic and spelling corrections are in red.

Reviewer #1 (Remarks to the Author):

In the paper titled "Long QT syndrome type 3 gain-of-function of Nav1.5 increases ventricular fibroblasts proliferation and pro-fibrotic factors" by Castro et al, based on their previous study and combined with the clue of expression of SCN5A gene in cardiac fibroblasts, the author proposes a scientific question "the potential role of gain of function of Nav1.5 in cardiac fibroblasts". The results show that cardiac fibroblasts from Scn5a+/ Δ QKP knock-in mice have stronger proliferation ability compared to the control mice, and this high proliferation ability is related to the activation of TGF signaling pathway. In addition, abnormal calcium signal activity could partially explain the high proliferation activity of cardiac fibroblasts from the Scn5a+/ Δ QKP knock-in mice. This is an interesting question to be addressed because the role of this gene in the cardiac fibroblasts is still unknown. In this sense this paper adds a contribution to these findings. Before considering acceptance, some experiments need to be supplemented and the following concerns need to be addressed.

Major concerns:

1. The author used 4-week-old mice in the experiment. In addition to observation of obvious proliferation of cardiac fibroblasts in the +/ Δ QKP group, there was also a significant phenotype of cardiac enlargement. It is necessary to provide experimental evidence to prove whether this enlargement is caused by an increased number of fibroblasts or hypertrophy of myocardial cells. Considering the important role of fibroblasts in myocardial hypertrophy, it is necessary to conduct the same experiment at more time points (such as Fig1), such as 1 week before birth, 1 and 2 weeks after birth. The dynamic data presentation is of great significance for readers to obtain more information and clues.

As mentioned in the first paragraph of the results section, in a previous study, we had shown (reference 5 of the current manuscript) that there was indeed a cardiomyocyte hypertrophy at the age of 4 weeks that could explain the larger left ventricular wall and septum thickness, as well as the larger heart weight / tibia length ratio in Scn5a+/ Δ QKP mice compared to control mice (see figure 2, supplemental table 1 and supplemental figure 5 of reference 5). I was not our intention here to state that cardiac hypertrophy was exclusively due to fibroblast proliferation. To avoid such misunderstanding, we have now modified the first paragraph of the results section (page 4, lines 14-15) by adding the following sentence: "This increase in heart size was attributed to increased left ventricular wall and septum thickness due, at least in part, to cardiomyocyte hypertrophy."

In our previous study (reference 5), there was no evidence for cardiac hypertrophy in 2-week-old mice. Following the reviewer's advice, we have performed additional experiments at the age of two weeks. Our results show that at this age, there was no difference in vimentin staining between control and Scn5a+/ Δ QKP mice (new extended data Fig. 2), while the proportion of cells stained with Ki67 tended to increase (non-significant) in Scn5a+/ Δ QKP mouse hearts. This suggests that the age of two weeks corresponds to the onset of fibroblast proliferation. The first paragraph of the results section has been extended as follows (page 4, lines 21-27): "However, similar vimentin expression was measured in isolated fibroblasts from the two mouse types (Fig. 1d), thus confirming the increase in fibroblast abundance in Scn5a+/ Δ QKP hearts at the age of 4 weeks. In contrast, in cardiac sections from 2-week-old mice, fibroblast abundance did not differ between Scn5a+/ Δ QKP and control mice, as shown by

vimentin staining experiments (Extended Data Fig. 2). However, the slight non-significant increase in cell proliferation, as shown by Ki67 signal measurement (Extended Data Fig. 2), suggests that the age of two weeks corresponds to the onset of fibroblast proliferation."

2. Although the author mentioned some seemingly contradictory aspects regarding fibroblast activation in the discussion and proposed a hypothesis that the fibroblasts in this study stimulated by gain of function of Nav1.5 may be at the stage of proto-fibroblasts, there are indeed some deviations from consensus in the results presented. For example: *Scn5a*^{+/ Δ QKP} mice have high levels of collagen secretion, high protein levels of CTCF and inflammatory factors in the heart, while α -SMA expression is down-regulated. It is strongly recommended that further and deeper discussion was added in the discussion section to discuss this point. In addition, some more experiments should be designed to prove the possible stage of proto-fibroblast, such as detecting specific proto-fibroblast-specific markers to preliminarily prove the hypothesis.

We thank reviewer 1 for this comment and suggestion to add discussion points regarding fibroblast activation. Unfortunately, there are currently no specific markers for proto-myofibroblasts, and their characterization is essentially based on phenotypic changes and the absence of expression of myofibroblast markers such as α -SMA. In the discussion, we expanded on current knowledge regarding the characterization of the state of proto-myofibroblasts versus myofibroblasts (page 12, lines 8-23):

" Our hypothesis is that the *Scn5a*^{+/ Δ QKP} fibroblasts could be at a proto-myofibroblast stage. Indeed, during their differentiation, fibroblasts undergo an intermediate stage of proto-myofibroblasts which proliferate, migrate and increase their production of ECM and cytokines before fully differentiating into myofibroblasts. This transitional and activated state is still poorly characterized.⁸⁴ A known feature of proto-myofibroblasts is a rearrangement of the actin cytoskeleton with the formation of stress fibers that extend along the length of these cells. The absence of α -SMA within these stress fibers distinguishes proto-myofibroblasts from myofibroblasts which do exhibit α -SMA-positive stress fibers. Moreover, myofibroblasts, in contrast to proto-myofibroblasts, are characterized by low migration and proliferation capacities, and increased contractility^{85,86}. The increased proliferation rate, ECM production and formation of stress fibers, with a low expression of α -SMA, in *Scn5a*^{+/ Δ QKP} fibroblasts support our hypothesis that they are proto-myofibroblasts^{15,87,88}. Whatever the case, it is widely acknowledged that the majority of cardiac fibroblasts form via epithelial-to-mesenchymal transition⁸⁹, and Nav1.5 has been shown to regulate Epithelial-to-Mesenchymal Transition (EMT) and invasiveness in breast cancer cells⁹⁰. Further studies are warranted to ascertain whether Nav1.5 is also involved in the transition between fibroblasts, proto-myofibroblasts, and myofibroblasts, as suggested by our findings. "

Five references on this subject of discussion have been added:

84. Tai, Y. *et al.* Myofibroblasts: Function, formation, and scope of molecular therapies for skin fibrosis. *Biomolecules* **11**, 1–27 (2021).
85. D'Urso, M. & Kurniawan, N. A. Mechanical and Physical Regulation of Fibroblast–Myofibroblast Transition: From Cellular Mechanoresponse to Tissue Pathology. *Front. Bioeng. Biotechnol.* **8**, 1–15 (2020).
86. Correa-Gallegos, D. *et al.* CD201+ fascia progenitors choreograph injury repair. *Nature* **623**, (2023).
89. Plikus, M.V. *et al.* Fibroblasts: Origins, definitions, and functions in health and disease. *Cell*. **15**, 3852-3872. (2021)
90. Gradek, F. *et al.* Sodium Channel Nav1.5 Controls Epithelial-to-Mesenchymal Transition and Invasiveness in Breast Cancer Cells Through its Regulation by the Salt-Inducible Kinase-1. *Scientific Reports*. **9**, 8652 (2019).

3. It is inferred that homozygous mice with Δ QKP may be embryonic lethal. It will be quite interesting to investigate the effect of conditional expression of Δ QKP in the fibroblast-specific or inducible manner on the phenotype of mice.

We fully agree with the reviewer but believe that generating and investigating the relevant mouse models represents a large project by itself that is far beyond the subject of the present project.

4. The results of immunofluorescence staining of cardiac myocytes and fibroblasts in Fig-S8 and Fig1 need to be improved, with inconsistent sections of representative regions in which inconsistent orientation of cardiac myocytes (transverse vs longitudinal). It is recommended to perform whole-heart sections on the coronal section (it will display the four chambers of the heart), based on which the results are more comprehensive, objective, and accurate.

We agree with the reviewer that the orientation of cardiac myocytes on our immunostainings was inconsistent. We have solved this problem by providing for each mouse type a transverse and a longitudinal image. Legends of Fig.1 and Extended Data Fig. 10 (former 8) have been changed accordingly.

Regarding the request to perform whole-heart sections on the coronal section, we believe that this will not provide precision. Indeed, for histological staining such as fibrosis, a coronal section allows for an overall view of the areas of significant remodeling. Regarding staining with vimentin and alpha-actinin, as can be seen in the trials below, the use of low magnification to see the four cardiac chambers leads to a loss of staining precision (images A and B). It is necessary to use a higher magnification as we did in the article to visualize and quantify accurately the staining (image C). If fibrosis was patchy, showing whole heart coronal sections, as we did in the past for *Scn5a* heterozygous knockout mice (Leoni et al. PLoS One. 2010;5:e9298. doi: 10.1371/journal.pone.0009298), would be highly relevant, but it is less relevant for diffuse fibrosis, as in the present case.

Nucleus α -Actinin Vimentin

Minor concerns:

1. In Fig2B, IF detection of cell proliferation should also include markers for cardiac fibroblasts to prove which Ki67-positive cells belong to cardiac fibroblasts but not the other cell types. Similarly, whole-heart sections should be performed on the coronal section.

As already mentioned, we have very few specific markers for fibroblasts. Like most authors, we used vimentin, an intermediate filament. When cells enter mitosis to proliferate, the cytoskeleton is disrupted, and vimentin staining becomes diffuse. Under these conditions, it becomes impossible to obtain a quantifiable and reproducible co-staining of vimentin and Ki67. Our attempts have been unsuccessful. Additionally, regarding proliferating cells, it is improbable for proliferating cells to be cardiomyocytes, or even endothelial cells since Ki67 staining is not localized in vessels. Furthermore, to exclude proliferation of resident immune cells, we conducted immune cell markers staining with CD68. The results do not show any change in CD68 expression in the left ventricle. We have added

these results in a new Extended Data Figure 3. The results are presented page 5, lines 9-10 : “and our results on CD68 cardiac expression (Extended Data Fig. 3) suggest that the quantity of macrophages does not differ between control and *Scn5a*^{+/ Δ QKP} mice.”. As a consequence, Extended data figures have been renumbered.

Extended Data Fig. 3. Expression of CD68, a marker of macrophages, in *Scn5a*^{+/ Δ QKP} mouse ventricle is unchanged.

Representative western blot and CD68 expression in 4-week-old control and *Scn5a*^{+/ Δ QKP} mouse ventricular tissue (n = 12 & 8, respectively).

2. In Fig2C, in addition to Western Blot, detection of apoptosis should also include whole-heart sections on the coronal section using TUNNEL or IF of cleaved caspase 3.

We understand the reviewer's request. However, apoptosis in the heart is a rare event, and while it is feasible to demonstrate an increase, showing a decrease in apoptosis, as we did in fibroblast cultures, seems unrealistic in heart sections.

3. The author detect the protein level of SMAD2/3 using the method of Western Blot, and the result showed only one band. However, as the different family numbers of SMAD family, the result of Western Blot for SMAD2/3 should have two bands corresponding to SMAD2 and SMAD3. Please confirm the information of antibody used and address this issue.

We understand the reviewer's concern regarding the SMAD2/3 labelling. We indeed used an antibody detecting SMAD2/3, and we detected two bands on this Western blot. We have updated the examples of Western blots showing the phosphorylated form of SMAD2/3 (fig.3). Depending on the migration time used in the Western blots, the separation of the two bands varies. This explains why the two identified bands are very close on some Western blots.

Reviewer #2 (Remarks to the Author):

In this manuscript the authors extend their studies directed toward determining functional effects of sodium channel expression/activation in a cardiac fibroblast population isolated from the ventricles of adult mice engineered to express a mutant sodium channel that has been shown to have an enhanced late sodium current. The manuscript reports the phenotype of this fibroblast population in terms of markers for fibroblasts vs. myofibroblasts; the presence of TGF-beta receptors and its downstream signaling pathway; and some aspects of intracellular calcium signaling in these cells. After reading and then rereading this complex paper my assessment is that it presents a number of related and novel data sets and is directed toward an important translational pathophysiology. Unfortunately however, few if any of its main claims or conclusions are complete or convincing as presented. In revising and extending this work the authors will need to consider:

1. Whether their study can or should be described in terms of a sodium channel mutation with a sole emphasis on the alpha subunit of the Nav1.5 channel; or whether an extended study could or should also include analyses of the sodium channel complex - including the beta subunit composition in this study.

We agree with the reviewer that Nav1.5 is part of a larger complex, although it is the most important subunit, at least functionally since it generates a current close to native current when expressed in heterologous cell models. Understanding the effects of the alpha subunit of the channel is essential before looking at the effects of its regulatory subunits. Analyzing the whole sodium channel complex would make an already complex study even more complex. This could represent the subject of another study aimed at first identify the regulatory subunits expressed in fibroblasts, not limiting to β -subunits – there are regulatory subunits in cardiomyocytes that are at least as functionally important as them – and then identify their function by knocking-down their expression. In our opinion, this is far beyond the subject of the current study.

2. Adding information that would allow the reader/reviewer to have some idea of the sodium channel density in the isolated ventricular fibroblast preparation. In the absence of this it is not possible to relate the data sets generated by the indirect assays either to existing previously published data or to the functional ventricular substrate at baseline or in disease states.

We thank the reviewer for this comment. The expression of Nav1.5 was shown in Extended Figure 7d but was not properly referenced in the text. We have moved these Western blot results to Figure 4a and added quantifications of Nav1.5 expression.

3. Including more information concerning the actions of veratridine, including its concentration-dependent effects. It is unclear whether veratridine would be expected to enhance sodium current in a quiescent preparation such as the cardiac fibroblast, or even in a fibroblast preparation whose resting potential oscillates as described on page 7. In my experience, isolated cardiac fibroblasts in primary culture have a stable resting potential.

Veratridine affects the inactivation process of voltage-gated sodium channels, including Nav1.5, and dose-dependently increases the late current. More specifically, for Nav1.5, a concentration of about 8 μ M induces a late sodium current representing 10% of the peak current (Gulsevina et al. Int. J. Mol. Sci. 2022,23, 2225). At the concentration of 1 μ M used in our study, the increase in late current is more modest but corresponds to what is observed with the Δ QKP mutation, i.e., a late current representing 2-3% of the peak current (Wang DW et al. Proc Natl Acad Sci USA 1996;93(23):13200-5). This is now specified in the manuscript on page 7, lines 11-12: "A concentration of 1 μ M was chosen so that the late sodium represented 2-3% of the peak current, an effect comparable to the effect of the mutation." Actually, veratridine has already been shown to induce a late sodium current in non-excitable cells and alter their functional properties. For instance, veratridine has been shown to increase the late sodium current and dose-dependently enhance the invasiveness of cancer cells, an effect antagonized by tetrodotoxin (Gillet et al., 2009). It has also been shown to induce membrane potential oscillations in

non-excitabile glioma cells, resulting in a depolarization of the membrane potential, which is inhibited by tetrodotoxin (Reiser & Hamprecht, 1983). This is now specified in the discussion (page 9-10, lines 23-28 and 1-3) “Indeed, our results show that inhibiting this current with TTX or ranolazine decreases the proliferation of *Scn5a*^{+/ Δ QKP} mouse fibroblasts. In contrast, veratridine increases the proliferation of control fibroblasts. Veratridine has already been shown to induce a late sodium current in non-excitabile cells and alter their functional properties. For instance, veratridine has been shown to increase the late sodium current and dose-dependently enhance the invasiveness of cancer cells, an effect antagonized by tetrodotoxin⁴⁴. It has also been shown to induce membrane potential oscillations in non-excitabile glioma cells, resulting in a depolarization of the membrane potential, which is inhibited by tetrodotoxin⁴⁵. Thus, *in vitro*, activation of $\text{Na}_v1.5$ and changes in membrane potentials could be induced by veratridine.”

Concerning the reviewer's comment on our figure 5C and description on page 7, we respectfully point out that we did record intracellular calcium oscillations but not oscillations of the membrane potential. We cannot be sure that these calcium oscillations impacted the membrane potential, or that oscillations of the membrane potential induced intracellular oscillations, although membrane potential oscillations have already been observed in fibroblasts (Oiki S, Okada Y. Factors responsible for oscillations of membrane potential recorded with tight-seal-patch electrodes in mouse fibroblasts. *J Membr Biol.* 1988;105(1):23-32).

4. The section of the manuscript dealing with fibroblast calcium homeostasis is puzzling. Why was ATP at 100 micromolar chosen as the agonist and how does the putative activation of a number of different purinergic receptors relate to the main theme of this work?

We thank the reviewer for this comment which allows us to clarify our intention on this study. ATP, an agonist of purinergic (P2Y) receptors, is classically used as a chemical stimulation to study calcium signaling in fibroblasts. Fibroblasts express P2Y receptors which regulate the release of calcium from the endoplasmic reticulum (ER) via IP_3 -receptors by activating the $\text{G}\alpha_q/11$ protein – phospholipase C regulatory pathway (Feng et al., 2019; Lembong, Sabass, Sun, Rogers, & Stone, 2015; Lombardi et al., 2019). Our aim was not to connect $\text{Na}_v1.5$ to purinergic receptors but to use the activation of these receptors to trigger calcium release from ER in order to identify possible alterations of calcium homeostasis secondary to late sodium current-induced intracellular sodium overload. Our hypothesis was that alterations of calcium signaling is one of the processes downstream the increase in late sodium current in fibroblasts.

This hypothesis was based on studies performed on non-excitabile cells showing that $\text{Nav}1.5$ participates in the modification of calcium signaling resulting in the modulation of cellular processes such as proliferation (Andrikopoulos et al., 2011; Lo, Donermeyer, & Allen, 2012; Wang et al., 2000). In particular, the influx of sodium into cells via $\text{Nav}1.5$ channels has been shown to decrease the forward mode of NCX exchanger function and favor its reverse mode, thus favoring calcium accumulation in cells, as shown in astrocytes (Pappalardo, Samad, Black, & Waxman, 2014). Additionally, calcium signaling plays a central role in fibroblast function. For example, one study showed that Ca^{2+} signals activate the TGF- β /Smad signaling pathway and thus promote fibroblast proliferation and protein expression in the extracellular matrix (Liu et al., 2017).

We therefore investigated the $\text{Nav}1.5/\text{NCX}/\text{calcium}$ signaling pathway. We first showed that inhibition of the NCX exchanger with the inhibitor YM244769 (Yamashita, Watanabe, Kita, & Iwamoto, 2016), reduces the proliferation of fibroblasts from *Scn5a*^{+/ Δ QKP} mice, thus showing an involvement of the NCX exchanger in the LQT3 fibroblast phenotype. To go a step further, we investigated whether there are modifications of calcium signaling in cardiac fibroblasts. In our opinion, the best way to dynamically visualize calcium signaling in cultured fibroblasts and determine possible calcium overload or dynamic changes was to activate the P2Y receptors.

We added some precision on the following parts: in materiel and method (page 16, lines 5-7) “Adenosine triphosphate (ATP) was used as a traditional chemical stimulation to study calcium signaling in fibroblasts by inducing a calcium transient.^{92–94}” for the reason of the use of ATP, and in the results

(page 8, lines 23-27) “Finally, we investigated whether Ca²⁺ homeostasis was altered. Activation of purinergic receptors with adenosine triphosphate (ATP) was used to induce Ca²⁺ release from the endoplasmic reticulum via IP3 receptors, as classically performed.²⁷⁻²⁹ As shown in Fig. 5c, *Scn5a*^{+/ Δ QK^P ventricular fibroblasts had more frequent oscillations of intracellular Ca²⁺ and longer Ca²⁺ transients than control fibroblasts. “}

5. The new or revised manuscript needs to be edited and clarified extensively. There are a number of key sentences or sections that are ambiguous as a result of the use of unconventional Scientific English. Examples include:

- sentences in the Abstract on lines 8 and 9 and lines 12 and 13
- the last two sentences of paragraph 1 in the Introduction on page 3
- the entire last paragraph of the Introduction on page 4
- the text on lines 9 through 13 on page 9
- the section describing TGF-beta results that is on page 10 between lines 12 and 18
- the use of the word 'fibroblasts' throughout the manuscript. This should be the singular 'fibroblast' in almost all cases.

We thank the reviewer for raising these language problems. We have made numerous corrections throughout the manuscript (including the examples cited above). We hope that these corrections, that appear in red, will improve the clarity of the manuscript.

Comments:

In the paper titled “Long QT syndrome type 3 gain-of-function of Nav1.5 increases ventricular fibroblasts proliferation and pro-fibrotic factors” by Castro et al, based on their previous study and combined with the clue of expression of SCN5A gene in cardiac fibroblasts, the author proposes a scientific question “the potential role of gain of function of Nav1.5 in cardiac fibroblasts”. The results show that cardiac fibroblasts from Scn5a^{+/ Δ QKP} knock-in mice have stronger proliferation ability compared to the control mice, and this high proliferation ability is related to the activation of TGF signaling pathway. In addition, abnormal calcium signal activity could partially explain the high proliferation activity of cardiac fibroblasts from the Scn5a^{+/ Δ QKP} knock-in mice. This is an interesting question to be addressed because the role of this gene in the cardiac fibroblasts is still unknown. In this sense this paper adds a contribution to these findings. Before considering acceptance, some experiments need to be supplemented and the following concerns need to be addressed.

Major concerns:

1. The author used 4-week-old mice in the experiment. In addition to observation of obvious proliferation of cardiac fibroblasts in the +/ Δ QKP group, there was also a significant phenotype of cardiac enlargement. It is necessary to provide experimental evidence to prove whether this enlargement is caused by an increased number of fibroblasts or hypertrophy of myocardial cells. Considering the important role of fibroblasts in myocardial hypertrophy, it is necessary to conduct the same experiment at more time points (such as Fig1), such as 1 week before birth, 1 and 2 weeks after birth. The dynamic data presentation is of great significance for readers to obtain more information and clues.
2. Although the author mentioned some seemingly contradictory aspects regarding fibroblast activation in the discussion and proposed a hypothesis that

the fibroblasts in this study stimulated by gain of function of Nav1.5 may be at the stage of proto-fibroblasts, there are indeed some deviations from consensus in the results presented. For example: *Scn5a*^{+/ Δ QKP} mice have high levels of collagen secretion, high protein levels of CTCF and inflammatory factors in the heart, while α -SMA expression is down-regulated. It is strongly recommended that further and deeper discussion was added in the discussion section to discuss this point. In addition, some more experiments should be designed to prove the possible stage of proto-fibroblast, such as detecting specific proto-fibroblast-specific markers to preliminarily prove the hypothesis.

3. It is inferred that homozygous mice with Δ QKP may be embryonic lethal. It will be quite interesting to investigate the effect of conditional expression of Δ QKP in the fibroblast-specific or inducible manner on the phenotype of mice.

4. The results of immunofluorescence staining of cardiac myocytes and fibroblasts in Fig-S8 and Fig1 need to be improved, with inconsistent sections of representative regions in which inconsistent orientation of cardiac myocytes (transverse vs longitudinal). It is recommended to perform whole-heart sections on the coronal section (it will display the four chambers of the heart), based on which the results are more comprehensive, objective, and accurate.

Minor concerns:

1. In Fig2B, IF detection of cell proliferation should also include markers for cardiac fibroblasts to prove which Ki67-positive cells belong to cardiac fibroblasts but not the other cell types. Similarly, whole-heart sections should be performed on the coronal section.

2. In Fig2C, in addition to Western Blot, detection of apoptosis should also include whole-heart sections on the coronal section using TUNNE or IF of cleaved caspase 3.

3. The author detect the protein level of SMAD2/3 using the method of Western Blot, and the result showed only one band. However, as the different family numbers of SMAD family, the result of Western Blot for SMAD2/3 should have two bands corresponding to SMAD2 and SMAD3. Please confirm the

information of antibody used and address this issue.

4. In Fig2C, in addition to Western Blot, detection of apoptosis should also include whole-heart sections on the coronal section using TUNNE or IF of cleaved caspase 3.